# Selection of Animal Welfare Indicators for Primates in Rescue Centres Using the Delphi Method: *Cebus albifrons* as a Case Study

**DOI:** 10.3390/ani15172473

**Published:** 2025-08-22

**Authors:** Victoria Eugenia Pereira Bengoa, Xavier Manteca

**Affiliations:** 1Colombian Observatory of Animal Health and Welfare, Universidad de La Salle, Bogotá 110141, Colombia; 2Department of Animal and Food Science, School of Veterinary Science, Universitat Autònoma de Barcelona, 08193 Barcelona, Spain; 3Animal Welfare Education Centre (AWEC), Universitat Autònoma de Barcelona, 08193 Barcelona, Spain

**Keywords:** animal welfare, rescue centres, Delphi method, *Cebus albifrons*, welfare indicators, expert consensus

## Abstract

Wildlife rescue centres house a wide range of species, posing significant challenges to ensuring consistent standards of animal welfare. This study focused on the white-fronted capuchin monkey (*Cebus albifrons*) and developed two welfare assessment protocols: a daily monitoring checklist and a more comprehensive audit protocol. The Delphi technique—a widely used method for building expert consensus—was employed to identify, evaluate, and prioritise welfare indicators. A panel of 23 primate care experts participated in two Delphi rounds, leading to the validation of 28 indicators for the extended protocol and 10 for daily application. These included animal-based indicators—such as signs of pain, affiliative behaviours, and abnormal repetitive behaviours—as well as resource- and management-based indicators, including appropriate food provision, physical enrichment, and habitat dimensions. Although resource- and management-based indicators were more numerous and valued for their practicality in identifying underlying causes and risk factors, the highest-ranked indicators across both protocols were animal-based. These tools provide structured and adaptable strategies to support both daily care routines and drive long-term welfare improvements in rescue settings. Furthermore, they highlight the importance of integrating welfare assessment into conservation and rehabilitation efforts. Future research should focus on field-testing these tools, validating their reliability, and promoting evaluator consistency.

## 1. Introduction

Wildlife rescue and rehabilitation centres are facilities designed to temporarily house terrestrial and aquatic wildlife specimens that have been seized, confiscated, voluntarily surrendered, or directly rescued from the wild for the purposes of care, evaluation, and treatment [1,2]. The animals’ temporary stay is maintained until a final decision regarding their disposition is made. This may include their release into the wild (reintroduction), relocation to an ex situ institution—such as a zoo, conservation centre, or research facility—or, in some cases, euthanasia [3,4]. These centres often accommodate a wide range of species from various taxonomic groups and geographic origins. The continual intake and release of animals require ongoing adjustments to infrastructure and conditions to address the biological and ecological needs of each species. Despite the crucial role these centres play, many operate with limited financial resources, which makes providing an adequate environment for the animals even more challenging [5,6].

The intake of primates into rescue centres is common, as this taxonomic order is among the most affected by both legal and illegal trafficking—primarily for use in biomedical research, the entertainment industry, or as pets [7,8,9,10]. *Cebus albifrons* (Humboldt, 1812) is a species of graceful capuchin monkey belonging to the family Cebidae. Its taxonomy has recently been revised, with previously recognised subspecies now elevated to species status: *Cebus albifrons* (Humboldt’s white-fronted capuchin), *Cebus leucocephalus* (Perijá white-fronted monkey), *Cebus malitiosus* (Santa Marta white-fronted capuchin), and *Cebus versicolor* (Varied white-fronted capuchin). *Cebus albifrons* is distributed across Colombia, Peru, Ecuador, Bolivia, Brazil, and Venezuela, at elevations ranging from sea level to 2000 m above sea level [7,11,12]. The species is classified as Least Concern (LC) by the International Union for Conservation of Nature (IUCN) Red List of Threatened Species [13], listed as Vulnerable (VU) in Colombia [14], and included in Appendix II of the Convention on International Trade in Endangered Species of Wild Fauna and Flora (CITES).

The welfare of these animals in captivity is significantly influenced by the conditions they experience following capture and during their transfer to rescue centres. Animal welfare is defined as “the physical and mental state of an animal in relation to the conditions in which it lives and dies” [15,16]. It is unquestionably one of the most pressing concerns of our time, as it calls into question the consequences and impacts of both traditional and current practices in animal husbandry and production—wildlife in rescue centres being no exception.

The systematic assessment of animal welfare is fundamental for institutions responsible for ensuring it. Such evaluations enable the early detection of welfare issues, support the anticipation of adverse outcomes, facilitate effective risk analysis, and allow for the timely implementation of appropriate interventions [17,18]. They play a vital role in preventing unnecessary suffering, promoting positive welfare states, and informing prompt decisions regarding the final disposition of animals—such as release—based on their welfare status.

Welfare assessments are typically based on three main types of indicators: animal-based, resource-based, and management-based. Each type of measure offers distinct advantages and faces specific limitations; therefore, a comprehensive approach that integrates multiple indicators is widely recommended [19]. Animal-based measures, which require direct observation [20], provide the most immediate insight into an individual’s condition. These include behavioural observations—such as signs of pain, fear, or abnormal repetitive behaviours—physiological parameters, such as glucocorticoid levels, glucose concentration, or immune response, and clinical health indicators, including body condition and lameness. Resource-based indicators focus on the animals’ physical and environmental conditions. Examples include access to essential resources such as food, water, and shelter, as well as enclosure size, available space, ambient temperature, and the provision of bedding materials [21]. Management-based indicators refer to the operational practices within the institution. These include veterinary care, the quality of human–animal interactions, adherence to standard operating procedures, and the maintenance of health records.

To be considered robust, any welfare indicator must be evaluated in terms of its validity, reliability, and practical applicability. Ideally, a welfare measure should optimise all three attributes. However, in practice, reliability is often prioritised, as an unreliable measure undermines the ability to draw meaningful conclusions [22]. The most relevant forms of reliability in welfare assessment are inter-observer reliability, intra-observer reliability, and test–retest reliability, all of which ensure that findings are not dependent on who conducts the evaluation or when it is conducted, assuming stable conditions [23,24]. Inter-observer reliability refers to the consistency of measurements across different observers, whereas intra-observer reliability assesses the consistency of a single observer when repeating the evaluation. Test–retest reliability, by contrast, focuses on the temporal stability of a measure—whether the same result can be obtained when the assessment is repeated at a different time under similar conditions. This form of reliability is particularly useful for verifying that an indicator remains stable and interpretable over time, and is not influenced by external fluctuations or learning effects [25,26]. Validity, on the other hand, concerns how accurately a measure reflects the specific attribute it is intended to assess [20]. Valid measures reliably represent the underlying welfare state, are specific to the targeted aspect, and align with the relevant biological or behavioural phenomena [23]. Validity can be further categorised into content validity (expert consensus), construct validity (representation of theoretical concepts), and criterion validity (correlation with a recognised benchmark). In animal welfare science, construct validity is often emphasised due to the absence of universally accepted gold standards in many areas of assessment [22]. Finally, the practical applicability of any measure is crucial for its integration into routine welfare assessment protocols. Scientifically sound indicators may become unfeasible if their implementation requires excessive time, financial resources, personnel, or training. Therefore, practicality—understood as the balance between scientific rigour and operational feasibility—is essential to ensure the long-term sustainability and effectiveness of animal welfare assessment programmes.

To define, prioritise, and harmonise animal welfare indicators, the Delphi method offers a robust and widely recognised approach. It is a structured, iterative process designed to gather and synthesise expert opinion through successive rounds of questionnaires, accompanied by controlled feedback, while maintaining participant anonymity throughout the process [27,28,29]. This method is particularly valuable when involving experts from diverse backgrounds, levels of experience, or areas of specialisation, and in contexts where direct communication may be limited or unfeasible. It enables participation from a broader range of individuals than would typically be feasible in face-to-face meetings—particularly under time or budgetary constraints—and helps to address politically sensitive or contested issues by ensuring impartial input through anonymity. Moreover, it preserves participant heterogeneity, reducing the risk of dominance by more assertive individuals or the emergence of a bandwagon effect, thereby enhancing the validity of the results [30].

Assessing welfare through objective, scientific, and specific protocols or systems in wildlife rescue centres constitutes both an ethical and moral responsibility. At present, no existing system, method, or protocol provides valid, reliable, and practical indicators for evaluating the welfare of *Cebus albifrons* or other neotropical primates in such settings. This gap highlights the need for tools specifically adapted to the realities of rescue centres, where animals’ histories, housing, and handling conditions differ significantly from those found in more permanent and stable environments such as zoos and bioparks. This study aimed to evaluate the content validity of proposed welfare indicators through a Delphi consultation process and to develop two welfare assessment protocols for *Cebus albifrons*: one intended for routine use by keepers, and another designed for more comprehensive evaluations conducted by professionals.

## 2. Materials and Methods

### 2.1. Phases of the Study

#### 2.1.1. Delphi Procedure

The Delphi method was implemented as outlined below: Figure 1.

#### 2.1.2. Initial Indicators and Expert Panel

Based on a review of the scientific literature and prior informal welfare assessments, 39 initial indicators were identified and classified as either direct (animal-based) or indirect (resource- and management-based) measures.

Experts were defined as individuals with experience in the handling, care, or treatment of primates—particularly *Cebus albifrons*—within rescue or rehabilitation centres, or those with expertise in animal welfare science. Experts were selected through existing contacts in wildlife rescue, as well as through online searches for relevant centres within the species’ geographic range. They were contacted via telephone, WhatsApp, and email to explain the project’s objectives and methodology and to confirm their willingness to participate in the study.

A total of 23 experts completed both rounds of the Delphi process. Of these, 22 were based in Colombia and one in Ecuador. Among them, 60.87% (*n* = 14) were veterinarians, 13.04% (*n* = 3) animal scientists, 8.07% (*n* = 2) researchers, and 14.39% (*n* = 4) held other roles, such as biologists, primatologists, or provided logistical and technical support. All experts held a university-level qualification. Specifically, 30.43% (*n* = 7) held an undergraduate degree, 13.04% (*n* = 3) a specialisation, 39.13% (*n* = 9) a master’s degree, 13.04% (*n* = 3) a doctoral degree, and 4.34% (*n* = 1) a postdoctoral qualification. The majority (86.96%) had experience working with primates, either under human care (52.17%) or in both in situ and ex situ contexts (34.78%). Of these, 41.83% had over 10 years of experience. Experts without direct primate experience worked in the field of animal welfare.

#### 2.1.3. Data Collection

The design, administration, and partial data analysis were conducted using the Qualtrics ^XM ®^ online platform. At the outset, key concepts—such as validity, practicality, and confidentiality—were explained, along with a definition of rescue centres. Two theoretical scenarios for welfare assessment were presented: the first entailed a long, audit-type protocol (Scenario One), while the second involved a shorter, daily or rapid evaluation protocol (Scenario Two) (Table 1).

#### 2.1.4. Pilot Test and First Delphi Round

Once the questionnaire had been developed, it was sent to five individuals with experience in animal welfare and survey design. They were asked to assess the types of questions, clarity, response options, and the time required to complete the questionnaire. Feedback from this pilot test was used to make the necessary adjustments.

After the final version of the questionnaire was prepared, a personalised link was sent via email to 82 selected experts. The survey remained open for two weeks. The questionnaire consisted of five sections. It began with an informed consent form, followed by a section collecting demographic data. Scenario One was then presented, comprising a matrix of animal-based and indirect indicators. Experts were asked to assess each indicator in terms of validity, reliability, and practicality, and to select the 10 indicators they considered most relevant. Scenario Two, structured similarly to Scenario One, was presented next. Again, experts were asked to assess and select the 10 most important indicators based on the same criteria. In the final section, participants were invited to suggest any additional indicators not previously included and to provide comments or observations.

#### 2.1.5. Second Delphi Round

The second questionnaire was distributed to the experts who had completed the first round. As in the previous stage, the survey remained open for two weeks and was divided into several sections. In the first section, indicators from Scenario One that had achieved a consensus of 70% or more across the three main criteria—validity, reliability, and practicality—were presented. Experts were asked to evaluate these indicators, which were organised in a matrix using the same structure as in the previous round, and to select the 10 indicators they considered most relevant. In the following section, a separate matrix included the new indicators proposed by experts during the first round. Experts were again asked to select the 10 indicators they deemed most important. The same process was applied to Scenario Two, where indicators that had reached a 70% consensus in the first round were reassessed. Experts were invited to adjust the ranking of the indicators or propose modifications, again considering the criteria of validity, reliability, and practicality. The questionnaire was accompanied by a document providing detailed feedback on the previous round, including the consensus percentage for each indicator and a summary of expert responses. The survey remained open for two weeks.

### 2.2. Welfare Assessment Protocols

The long-form welfare assessment protocol was developed using all indicators that achieved a consensus of 70% or more across the three criteria—validity, reliability, and practicality—during the second round of the Delphi process. In contrast, the daily protocol was constructed using the 10 indicators that not only met the consensus threshold but also received the highest importance rankings based on the validity criterion established by the experts. This ranking was determined by the mean frequency of importance assigned across both Delphi rounds.

### 2.3. Data Analysis

Descriptive statistics (frequencies, means, and medians), along with percentage measures, were used as the primary indicators of consensus. An a priori agreement threshold of 70% or higher was established, in line with previous studies on health and animal welfare in which indicators are validated or prioritised [29,31,32]. Incomplete questionnaires from the first round of the Delphi process were included if they contained complete information for at least one scenario. To avoid inflating the expert sample size, surveys that were initiated but not completed, or those submitted more than once by the same expert, were excluded. The data were imported into Microsoft Excel for Microsoft 365 and summarised for analysis. To preserve participant anonymity, all identifiers were removed. Qualitative analysis was performed on open-text comments. In the second Delphi round, consensus was also assessed using the coefficient of variation (CV), a standardised measure of dispersion useful for comparing distributions [19]. The CV was calculated using the following formula: CV = Standard Deviation / Mean Interpretation followed the framework outlined by English and Kernan [33] and subsequently adopted by Berteselli et al. [34]: 0 ≤ CV ≤ 0.5 indicates a good degree of consensus (no additional round required); 0.5 < CV ≤ 0.8 indicates a less satisfactory degree of consensus (an additional round may be needed); CV > 0.8 indicates a low degree of consensus (an additional round is definitely required). To assess the internal consistency or reliability of the instrument, Cronbach’s alpha (α) coefficient was calculated using the SPSS statistical package v25. Alpha values range from 0 to 1, with higher values indicating greater internal consistency. Conventionally, values equal to or above 0.7 are considered acceptable, values above 0.8 are considered good, and values exceeding 0.9 are considered excellent. Values below 0.5, or approaching 0, are indicative of poor reliability.

## 3. Results

### 3.1. Indicators from the First Delphi Round

In Scenario One, out of the 39 initial indicators, 10 (25.64%) were eliminated during the first round for failing to meet the 70% consensus threshold in terms of validity, reliability, and practicality. Of these, 9 were direct indicators (from a total of 18 proposed), and 1 was indirect (from a total of 21). Additionally, experts proposed 23 new indicators—15 direct and 8 indirect—which were subsequently incorporated into the questionnaire for the second Delphi round. In Scenario Two, 14 out of the 39 initial indicators (35.89%) were excluded, comprising 9 animal-based and 5 indirect measures. Furthermore, the experts suggested 16 new indicators—10 animal-based and 6 indirect—which were included for analysis in the second Delphi round.

### 3.2. Indicators from the Second Delphi Round

#### 3.2.1. Scenario One: Extended Protocol

In the second round, 24 of the 52 indicators (46.15%) were excluded, with 16 out of 24 being animal-based and 8 out of 28 being indirect. Table 2 presents the validated indicators, along with their corresponding percentage scores for content validity, reliability, and practicality, as well as the coefficient of variation for each.

In addition, experts selected the 10 direct or indirect indicators they considered most important based on the validity criterion for Scenario One. As illustrated in Figure 2, evident health alterations, abnormal repetitive behaviours or stereotypies, body condition, and affiliative behaviours with conspecifics—all of which are animal-based—occupied the top four positions.

#### 3.2.2. Scenario Two: Daily Protocol

In this scenario, 21 out of the 41 indicators (51.21%) were eliminated during the second round, including 13 of 19 animal-based measures and 8 of 22 indirect indicators. Table 3 displays the validated indicators, together with their percentage scores for validity, reliability, and practicality, as well as their coefficient of variation. Experts also selected the 10 direct or indirect indicators they deemed most important according to the validity criterion for this scenario. Three behavioural indicators ranked highest in terms of importance: abnormal repetitive behaviours or stereotypies, affiliative behaviours with conspecifics, and signs of pain (Figure 3).

### 3.3. Welfare Assessment Protocols for Cebus albifrons in Rescue Centres

#### 3.3.1. Extended Audit-Type Protocol

The implementation of the comprehensive audit-type protocol developed for Scenario One requires additional time, specialised veterinary and technical equipment, and, in some cases, further resources to perform physical and clinical examinations of the animals. This protocol includes 28 indicators (Table 4): 4 related to the principle of good nutrition, 12 to good housing, 3 to good health, 7 to appropriate behaviour, and 1 concerning staff/human resources. Of these, 8 are animal-based indicators, 13 are resource-based, and 7 are management-based.

#### 3.3.2. Daily Protocol

The daily protocol consisted of 10 out of the 20 indicators that achieved expert consensus, providing a more practical and manageable tool for routine implementation. These 10 indicators were selected based on two main criteria: (1) reaching at least 70% consensus for content validity, reliability, and practicality, and (2) being ranked highest in importance by the experts according to the validity criterion. The final assessment tool includes two indicators under the principle of good nutrition, three under good housing, one under good health, and four under appropriate behaviour. This streamlined approach was designed to minimise time and resource demands while maintaining effectiveness in assessing animal welfare (Table 5).

## 4. Discussion

The objectives of this study were, first, to validate welfare indicators for white-fronted capuchin monkeys (*Cebus albifrons*) through expert consensus, and second, to develop two welfare assessment protocols—one for daily use and the other more comprehensive—for implementation in rescue centres.

The Delphi method employed in this study has previously been used to identify welfare concerns and to define, prioritise, and harmonise indicators across a range of species, including dogs [34,35,36], cattle [37,38], poultry [39], sheep [40], pigs [37], horses [41,42,43], rabbits [44], mice [31], rats [43], and farmed fish [45]. It has also been applied to wildlife species such as reptiles from the families Agamidae, Chelidae, Pythonidae, Testudinidae [46], and Crocodylidae [47], as well as to tigers [48], elephants [49], stranded cetaceans [50], research primates [29], and parrots [51,52]. This technique facilitates structured group communication. One of the main reasons for its widespread use is that it enables the inclusion of numerous participants from diverse geographic locations and areas of expertise while preserving anonymity—thus preventing individual experts from dominating the consensus process [53]. Although there are no universally standardised requirements for the Delphi method, this study adhered to the guidelines proposed by Boulkedid et al. [54].

The first questionnaire was distributed to 82 experts who had agreed to participate. Of these, 42 experts responded to Scenario One and 37 to Scenario Two, yielding response rates of 51.22% and 45.12%, respectively. The second-round Delphi questionnaire was sent to the 42 experts who had participated in the first round, and 23 of them completed it (response rate: 54.76%). The internal consistency (reliability) of the instrument was excellent, as evidenced by a Cronbach’s alpha of 0.968. Although the literature does not offer definitive guidelines on minimum acceptable response or dropout rates, the response rate observed in this study is considered acceptable. Previous studies have reported dropout rates between Delphi rounds of up to 78% [46]. Declines in response rates between rounds may be attributable to various factors, including the total number of rounds, the time commitment involved (e.g., several weeks or months of engagement), distractions between rounds, or participant fatigue due to perceived repetitiveness or length of the questionnaires [55]. In this study, some experts may have lost interest or experienced frustration, particularly given that the survey comprised two scenarios, each subdivided into sections for direct and indirect indicators [46]. This may also account for the five-response discrepancy between Scenario One and Scenario Two in the first round. A third Delphi round was deemed unnecessary, as the coefficient of variation (CV) indicated a good degree of consensus (0 ≤ CV ≤ 0.5) for all indicators across the three criteria. To minimise the risk of declining participation, experts were encouraged to complete all rounds by highlighting the importance of their contribution in developing standardised protocols for assessing the welfare of white-fronted capuchin monkeys [34]. The deadline for each survey round was extended from two to three weeks, and personalised reminders were sent via email and WhatsApp prior to the closing date. With regard to the required number of panel experts, Turoff [56] recommends a minimum of 7 to 15 participants, and several studies have involved panels of between 15 and 39 experts [32,46,57]. Therefore, the panel of 23 experts in this study exceeded the minimum threshold. Moreover, the number of experts in primate welfare working in rescue centres may be greater than in other contexts, such as laboratory or research settings.

### 4.1. Extended Audit-Type Protocol—Scenario One

Across the two rounds of the Delphi method for Scenario One, 62 indicators were evaluated by experts, of which 32 (51.61%) were direct animal-based indicators and 30 (48.39%) were indirect indicators based on resources and management. Of the total measures discarded during the Delphi process, 79.59% were animal-based, and 47.06% were related to health. By the end of the process, 28 indicators (45.16%) achieved the 70% consensus threshold for validity, reliability, and practicality. This protocol may require extended observation periods, more intensive animal handling, and veterinary assessments. It incorporates measures aligned with the four principles of the Welfare Quality^®^ project; however, there is a marked predominance of resource- and management-based measures (20 out of 28). The experts’ tendency to favour these types of measures over animal-based ones may stem from the perception that they are more objective for welfare assessment and easier to implement in a wildlife rescue centre context. Such measures require minimal training to ensure high inter-observer reliability and can be evaluated quickly, often through numerical scales or binary assessments (e.g., presence or absence). Furthermore, they tend to remain stable over time and, unlike animal-based measures, are less influenced by variables such as time of day or weather conditions [58]. Assessing the resources available to animals can help identify the root causes of potential welfare issues, thereby contributing to their resolution [59,60]. Nonetheless, the mere provision of adequate resources does not necessarily ensure good welfare outcomes. Resource-based indicators do not directly assess the condition of the animals (i.e., the outcomes themselves), but rather serve as indirect proxies for animal welfare [37]. For instance, evaluating the number of feeders may indicate whether an enclosure or cage housing a group of *Cebus albifrons* provides a sufficient quantity for the group’s size, and may serve as an indicator for the absence of prolonged hunger. However, some individuals may still experience hunger if, for example, dominant monkeys monopolise the feeders or if competition limits access for others.

### 4.2. Principle of Good Feeding

The principle of good feeding addresses one of the fundamental physiological and behavioural needs of animals: access to adequate and appropriate food. In wildlife rescue centres, managing food provision poses particular challenges due to the wide range of species under care and their diverse nutritional requirements. These challenges are further compounded by the difficulty in providing naturalistic diets, which are essential not only for meeting species-specific nutritional needs but also for supporting the rehabilitation process by encouraging natural foraging behaviours. Experts validated four reliable and practical indicators under the criteria of absence of prolonged hunger and absence of prolonged thirst. Among these, the body condition score was rated highly for both validity and practicality (91.30%), likely due to its standing as one of the most robust, widely accepted, and preferred direct measures for assessing nutritional status over the medium and long term. This indicator is extensively used in welfare assessment protocols for production species [61,62,63,64,65], wildlife [66], and shelter dogs [67]. Although scoring systems have been proposed for other primates, such as macaques [68], a validated scale for New World primates does not currently exist, which may account for the moderate reliability score obtained for this measure (73.91%). Within this principle, the quantity, frequency, variety, and nutritional composition of the diet received the highest scores for validity (95.65%), reliability (91.30%), and practicality (91.30%). Nutritional imbalances are common in New World primates in captivity, often involving deficiencies in vitamins D_3_, A, and C, as well as calcium—deficiencies that can significantly affect tissue integrity, disease susceptibility, and body weight [69,70]. The number and cleanliness of feeders were identified as key short-term indicators of good nutrition (validity: 91.30%, reliability: 73.91%, practicality: 78.26%). It is crucial that food and containers are easily accessible to all primates within the enclosure, as this can be a determining factor in reducing conflict and aggression, and in ensuring equal access to food among individuals. Feeders should be cleaned at least once daily [69], a practice essential for maintaining hygiene and preventing potential health issues. Under the criterion of absence of prolonged thirst, the number of water points in the enclosure was prioritised by experts, scoring 91.30% for validity, 73.91% for reliability, and 78.26% for practicality. This indicator was favoured over others—such as the condition and cleanliness of water points, water consumption, and quantity and quality of drinking water—which were ultimately discarded during the process.

### 4.3. Principle of Good Housing

A total of 12 indicators under the principle of good housing were content validated for inclusion in the protocol, categorised into the following criteria: comfort during rest, thermal comfort, and ease of movement. Housing is one of the most critical factors in wildlife rescue centres, as it demands considerable staff effort and resources to adapt infrastructure to the specific needs of each species. With a validity score exceeding 86.96%, these measures demonstrate strong relevance for assessing animal welfare. Within the comfort during rest criterion, the presence of visual barriers both within and between enclosures is crucial. These barriers allow primates to choose whether to observe or hide from conspecifics and humans, granting them greater control over their social interactions. This contributes to the reduction of intra- and inter-group aggression and facilitates successful pairings and group formations [71]. The availability and cleanliness of shelters are particularly significant in enclosures that lack both indoor and outdoor areas, as shelters provide safe and suitable conditions for nocturnal rest. Under thermal comfort, the indicator enclosure with indoor and outdoor areas assesses whether primates have the opportunity to regulate their thermal environment. Indoor spaces are essential for protection from extreme weather, which is particularly relevant given that New World primates may be sensitive to low temperatures. Such areas also provide shelter from predators, aggressive conspecifics, and certain pathogens [72]. Outdoor access is equally important, offering exposure to sunlight—critical for vitamin D synthesis—as well as shade, rainfall, and fresh air, all of which support health and physiological well-being [73,74]. Animal density (validity: 100%, reliability: 95.65%, practicality: 100%) is a critical factor in wildlife rescue centres, particularly given that areas and enclosures or cages designated for housing capuchins are often limited. This constraint frequently necessitates the inclusion of additional individuals within a group, thereby increasing animal density. Elevated density levels can have significant consequences for animal welfare, primarily due to intensified competition for resources and heightened aggression, especially when enclosure structures do not provide primates with the means to avoid or escape from one another [69]. Forced proximity within confined spaces can generate a highly stressful environment, particularly for species with complex social dynamics such as primates. Social stress may manifest through aggressive behaviours, frequent conflicts, and the disruption of stable social hierarchies. Prolonged exposure to such social tension can adversely affect psychological well-being and may result in physical repercussions, including weakened immune function and increased susceptibility to disease [75,76]. Maintaining appropriate animal density within enclosures is thus essential to mitigate these risks. It fosters a more balanced and healthier environment, facilitates more natural social interactions, reduces social tension, and helps to prevent the emergence of abnormal behaviours. The indicator enclosure dimensions received validity, reliability, and practicality scores of 100%, 95.65%, and 91.30%, respectively. Currently, no specific standards exist for enclosure or cage dimensions for primates housed in rescue centres. Nevertheless, zoo management guidelines recommend a minimum height of 3 metres and a surface area of 40 m^2^ for a group of 10 individuals [72]. Enclosure dimensions are vital, as the space allocated to animals directly influences their ability to express species-specific behaviours, determines the feasible group size, and dictates the scope for environmental enrichment. Restricted enclosures constrain animals’ locomotor activity and limit opportunities for physical exercise, potentially resulting in muscle atrophy and reduced joint mobility [77,78]. Moreover, the incidence of stereotypic and other abnormal behaviours is often higher in smaller enclosures compared to larger, more structurally complex ones [79]. Vertical space—specifically, the presence of at least three levels—is particularly important, as cebid primates typically exhibit escape responses by moving upward, and dominance hierarchies are often expressed through occupation of elevated perches [74]. Access to multiple vertical levels enables individuals to select higher positions, granting them greater environmental control and opportunities to avoid social conflict, thereby contributing to enhanced group stability. The availability of isolation cubicles or dual-compartment enclosures is also essential. These features support low-stress animal handling and enhance safety for both animals and staff performing routine tasks such as cleaning, enrichment provision, and feeding. Finally, all indicators pertaining to environmental parameters—such as noise levels, lighting, temperature, and humidity within indoor enclosures—were excluded from this study owing to their low perceived reliability and practicality.

### 4.4. Principle of Good Health

Under the principle of good health, only three indicators exceeded the 70% threshold for content validity, reliability, and practicality—all of which fell under the criterion of absence of disease. One of these, evident health alterations, is considered crucial for the timely detection of health problems in *Cebus albifrons* housed in rescue centres, as it enables the implementation of appropriate therapeutic interventions and the mitigation of negative affective states associated with illness. It is essential that veterinarians possess the necessary skills and experience to recognise signs of illness, particularly given that primates often conceal such symptoms [69]. Another validated indicator is the evidence of a preventive medicine programme, which may include adherence to quarantine periods established for this taxonomic order in rescue facilities, veterinary assessments at key stages of their stay (e.g., upon admission, post quarantine, and prior to final disposition), faecal examinations, and other diagnostic tests for infectious diseases (primarily at intake and prior to release), as well as vaccinations and dental prophylaxis where appropriate. Lastly, mortality was also considered a relevant welfare indicator by the panel of experts, with a validity rating of 86.96% and practicality of 78.26%, given that the recording of this information is mandatory in rescue centres. All indicators associated with the absence of injuries—such as lameness, digit alterations, injuries caused by enclosures or cages, and those inflicted by conspecifics—were discarded. Similarly, other clinical signs indicative of disease, including alopecia score, ocular discharge, nasal discharge, vulvar discharge, dental or oral deterioration, and signs of pain, were eliminated early in the validation process.

### 4.5. Principle of Appropriate Behaviour

The four behavioural measures were categorised under the criteria of expression of social behaviour and expression of other behaviours, as defined in the Welfare Quality^®^ framework. Affiliative behaviours with conspecifics stood out for their high validity (91.30%) and relevance, ranking fourth in importance as indicators of good welfare. All primates are inherently social species; therefore, their interactions with conspecifics are vital to their welfare [80,81]. Being part of a group provides essential stimulation, comfort, reassurance, pleasure, and opportunities to learn species-typical behaviours through observation, physical contact, and communication [82]. Play, resting in contact, grooming, embracing, carrying, and sitting together—among other interactions—comprise the most relevant behavioural repertoire for this species and contribute to group cohesion [83]. It may be assumed that, similar to *Sapajus* sp., social interaction is a fundamental need for *Cebus albifrons* [84]. These behaviours are particularly important in wildlife rescue centres, where their expression may serve as indicators of strengthening social structures during rehabilitation processes [85]. Behavioural patterns observed with appropriate frequency and intensity—such as foraging, locomotion, and resting—can indicate positive welfare. Among these, foraging is particularly noteworthy, as *Cebus albifrons* allocates a significant portion of its time budget to this activity. Given its central role in the species’ daily life, foraging should be actively promoted and facilitated in captivity [80]. Doing so not only fulfils an essential biological requirement but also helps maintain mental stimulation and physical activity—both key to overall well-being. Abnormal repetitive behaviours or stereotypies, which received some of the highest scores for validity (95.65%) and importance (ranked second), should be employed cautiously as indicators of current welfare [86]. These behaviours may persist even after the original stressor has been removed, potentially reflecting historical rather than current welfare status [87,88]. Examples include pacing, head-twisting, circling, floating limbs, rocking, and body-spinning—patterns often associated with prolonged confinement [89]. Their presence may suggest compromised mental health [90] and negative affective states, possibly leading to pessimistic judgement [91]. Other validated indicators of poor welfare included abnormal qualitative behaviours other than stereotypies (e.g., coprophagy), with scores of 91.30% for validity, 73.91% for reliability, and 86.96% for practicality. Exhibition of species-typical behaviours at atypical times or intensities (e.g., resting, stillness, or locomotion) was also validated (86.96%, 73.91%, and 78.26%, respectively). Additionally, consensus was achieved on all three criteria regarding the provision of physical, cognitive, and sensory enrichment. The aim of environmental enrichment (EE) is to address the biological needs of primates, stimulate their cognitive abilities, and prevent boredom. In rescue centres, EE also plays a key role in preparing individuals for successful reintegration into the wild by promoting behaviours such as locomotion, foraging, predator avoidance, social interaction, and overall physical conditioning [92,93].

During the expert consultation, measures related to human–animal relationships—such as aggression towards keepers and interactions with keepers—were discarded. This decision likely reflects the complexity of understanding and generalising the type of bond that should be established between keepers and white-fronted capuchin monkeys in a wildlife rescue centre. Although these animals may remain in captivity for an extended period, by the end of their rehabilitation process they must perceive humans as predators or threats to enhance their chances of survival upon release. Striking a balance between the necessary human interaction for proper care and the preservation of natural behaviours is, therefore, a critical challenge in such settings. Ultimately, the indicator staff training in animal welfare was validated, underscoring the importance experts place on institutional knowledge and the practical application of welfare principles. Adequate staff training is essential to ensure that animal welfare practices are implemented consistently and effectively, fostering an environment in which animals can truly thrive.

### 4.6. Scenario Two—Daily Protocol

In the context of Scenario Two, 55 indicators were evaluated over two rounds of the Delphi method. Of these, 27 (49.09%) were direct, based on animal observations, while 28 (50.90%) were indirect, focusing on resources and management. As in Scenario One, a greater number of animal-based indicators were discarded during the process, accounting for 60% of those eliminated. By the end of the second Delphi round, 20 indicators (36.36%) reached the 70% consensus threshold for validity, reliability, and practicality. The top ten were selected based on experts’ assessments of their importance. The daily welfare review is intended to be an agile and practical process, conducted through observations made during routine management tasks, requiring only a few additional minutes of attention. Similarly to the extended protocol, this daily protocol primarily consists of resource- and management-based measures (six out of ten indicators), covering all four welfare principles. Notably, nine of the ten indicators were also included in the extended protocol; the only exception was signs of pain. Capuchins housed in rescue centres may experience pain due to various conditions and pathologies. Although no specific pain assessment scale currently exists for New World primates, the signs described for Old World primates can serve as a useful reference. These signs include postures such as sitting, huddling, or crouching with the head tilted forward and arms crossed over the body; a sad facial expression with glazed eyes; reduced movement; decreased appetite and interest; lack of grooming and poor coat condition; avoidance of companions and staff; increased attention from cage mates; facial expressions, teeth grinding, restlessness, and tremors accompanied by vocalisations; as well as behaviours such as touching, squeezing, rubbing, or scratching the painful area, and changes in aggression towards caregivers [68,94]. Although indirect welfare indicators can be assessed quickly and easily, the four animal-based indicators (signs of pain, affiliative behaviours with conspecifics, abnormal repetitive behaviours, and abnormal behaviours other than stereotypies) require that caregivers are skilled in detecting subtle changes in posture, demeanour, expression, movement, and behaviour of the animals under their care [95]. The challenge lies in ensuring that behavioural observations—and their interpretation—are reliable and reproducible across different observers, as has been demonstrated in some zoological institutions where a high level of agreement has been achieved among caregivers in assessing welfare indicators [96].

### 4.7. Limitations

This study presents certain limitations, notably a restricted geographical representation, as the majority of consulted experts were based in Colombia. This may have introduced a degree of regional bias. Although experts from other Latin American countries where *Cebus albifrons* is also present in rescue centres were invited to participate, their involvement was limited. This constraint may have affected the generalisability of the findings across the species’ wider geographical distribution. Moreover, potential ambiguities in the phrasing of some indicators, or the inherent complexity of those indicators, may have influenced expert evaluations. These factors could have contributed to variability in interpretation, thereby impacting the level of consensus achieved.

## 5. Conclusions

Promoting animal welfare and fostering continuous improvements in the housing and management of animals under professional care represent major challenges for wildlife rescue centres. These challenges are compounded by the high turnover of animals—through intake, release, or transfer—and the wide taxonomic diversity received, each species possessing specific biological and behavioural requirements. Welfare assessment processes provide a valuable opportunity to develop evidence-based action plans that identify and prioritise interventions aimed at enhancing animal welfare.

This study contributed to the identification and prioritisation of valid, reliable, and practical indicators, facilitating their integration into welfare assessment tools for *Cebus albifrons.* These protocols have the potential to benefit thousands of graceful capuchins (*Cebus* genus) rescued from illegal trafficking and currently housed in rescue centres across Latin America. The research presented here marks a critical first step in the development of welfare assessment protocols within a context where such studies remain limited. Their implementation may not only enhance awareness of the importance of animal welfare but also help embed it as an institutional priority alongside conservation initiatives. To advance this line of work, further efforts are required to assess both the time demands and feasibility of implementation under field conditions. Additionally, formal reliability evaluations—encompassing intra- and inter-observer consistency, as well as test–retest reliability—should be conducted.

Further recommendations include promoting collaborative research to expand the knowledge base on species housed in rescue centres; advancing the development of both generic and species-specific welfare protocols; strengthening inter-institutional efforts to create standardised tools and coordinated responses to shared welfare challenges; and investing in capacity-building programmes to train staff responsible for animal care. The broader the adoption of these methodologies across rescue centres, the greater the sector’s overall capacity to promote animal welfare.

## Figures and Tables

**Figure 1 animals-15-02473-f001:**
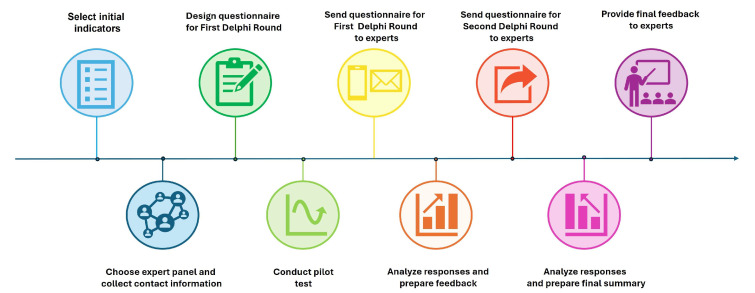
Step-by-step Delphi methodology applied in the study.

**Figure 2 animals-15-02473-f002:**
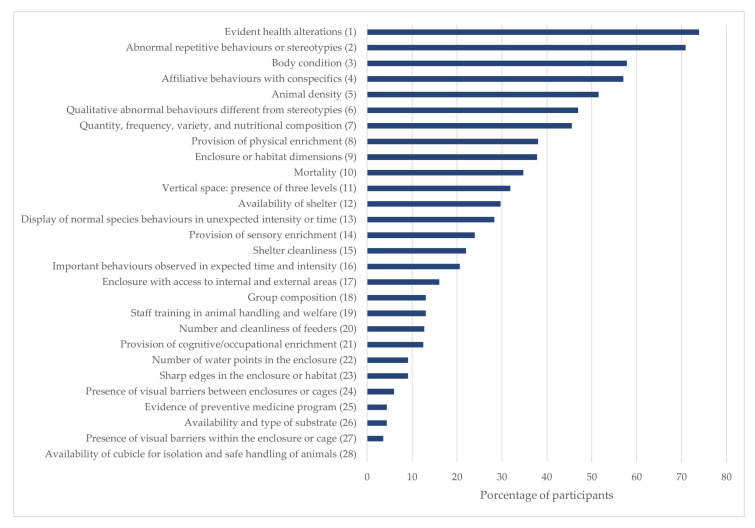
Ranking of indicators for Scenario One, based on the average frequency of the top 10 indicators selected across both Delphi rounds according to the content validity criterion.

**Figure 3 animals-15-02473-f003:**
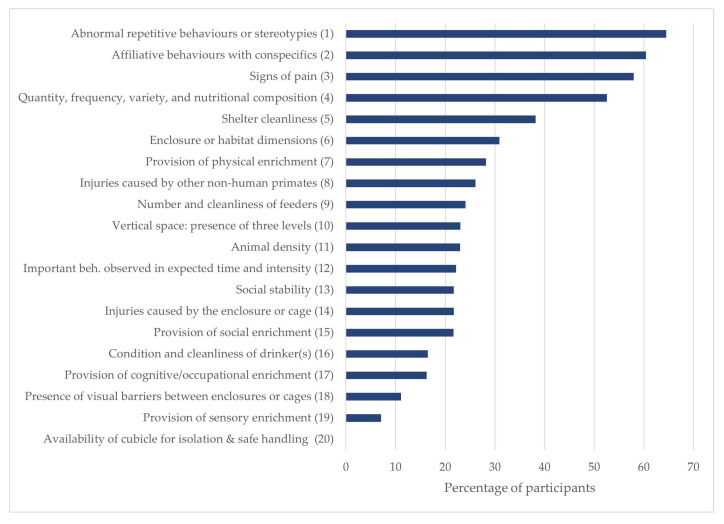
Ranking of indicators for Scenario Two derived from: (**a**) indicators that achieved at least 70% consensus on validity, reliability, and practicality; and (**b**) the average frequency of the top 10 indicators selected across both Delphi rounds, based on the content validity criterion.

**Table 1 animals-15-02473-t001:** Theoretical scenarios presented for welfare assessment.

Scenario One	Scenario Two
You are about to assess the welfare of 10 white-fronted capuchin primates (*Cebus albifrons*) that are housed in a wildlife rescue centre, with an estimated assessment time of eight hours. The individuals are housed either individually or in small groups in enclosures or maintenance cages, as they are not currently part of any group undergoing behavioural rehabilitation. You can request or perform physical and/or chemical restraint of the animals, and you have access to portable equipment (e.g., stethoscope, thermometer), questionnaires, and biosafety items such as gowns, masks, gloves, etc.	You are about to assess the welfare of 10 white-fronted capuchin primates (*Cebus albifrons*) that are under your care at the wildlife rescue centre during your daily rounds. The individuals are housed either individually or in small groups in enclosures or maintenance cages, as they are not currently part of any group undergoing behavioural rehabilitation. You have access to questionnaires and biosafety items such as overalls, masks, and gloves.

**Table 2 animals-15-02473-t002:** Indicators that reached 70% expert consensus in the second Delphi round, including percentage scores for Scenario One.

	Welfare Indicator	Content Validity (%)	CV	Reliability (%)	CV	Practicality (%)	CV
Animal-based	Body condition	91.3	0.1	73.91	0.2	91.3	0.1
	Affiliative behaviours with conspecifics	91.3	0.1	86.96	0.2	91.3	0.2
	Abnormal repetitive behaviours or stereotypies	95.65	0.1	91.3	0.2	91.3	0.2
	Abnormal qualitative behaviours different from stereotypies	91.3	0.1	73.91	0.2	86.96	0.2
	Important behaviours observed at expected time and intensity	82.61	0.1	73.91	0.2	78.26	0.2
	Normal species-specific behaviours at unexpected time/intensity	86.96	0.1	73.91	0.2	78.26	0.3
	Evident health alterations	100	0	91.3	0.1	91.3	0.2
	Mortality	86.96	0.2	82.61	0.2	78.26	0.2
	Quantity, frequency, variety, and nutritional composition	95.65	0.1	91.3	0.2	91.3	0.2
	Number of water points in the enclosure	91.3	0.2	73.91	0.3	91.3	0.1
	Number and cleanliness of feeders	91.3	0.1	73.91	0.2	78.26	0.1
	Sharp edges in the enclosure or habitat	95.65	0.1	73.91	0.2	82.61	0.2
	Animal density	100	0	95.65	0.1	100	0
	Enclosure or habitat dimensions	100	0	95.65	0.1	91.3	0.2
	Availability of shelter	91.3	0.1	78.26	0.1	95.65	0.1
	Enclosure with access to indoor and outdoor areas	95.65	0.1	78.26	0.1	86.96	0.2
	Vertical space: presence of three levels	86.96	0.1	78.26	0.1	86.96	0.2
	Cleanliness of shelter	91.3	0.1	78.26	0.2	86.96	0.2
Resource and management-based	Presence of visual barriers inside the enclosure or cage	91.3	0.1	73.91	0.2	86.96	0.1
	Presence of visual barriers between enclosures or cages	91.3	0.1	86.96	0.2	95.65	0.1
	Provision of physical enrichment	95.65	0.1	91.3	0.2	82.61	0.2
	Provision of sensory enrichment	86.96	0.2	78.26	0.2	78.26	0.2
	Provision of cognitive/occupational enrichment	91.3	0.2	82.61	0.2	73.91	0.3
	Staff training in animal management and welfare	86.96	0.2	78.26	0.3	78.26	0.2
	Group composition (age/sex)	82.61	0.1	82.61	0.2	82.61	0.2
	Availability and type of substrate	91.3	0.1	73.91	0.2	78.26	0.2
	Availability of cubicles for isolation and safe handling	91.3	0.2	86.96	0.2	78.26	0.2
	Evidence of preventive medicine programs	86.96	0.1	82.61	0.2	82.61	0.2

**Table 3 animals-15-02473-t003:** Indicators that reached 70% expert consensus in the second Delphi round, based on percentage scores for Scenario Two.

	Welfare Indicator	Content Validity (%)	CV	Reliability (%)	CV	Practicality (%)	CV
Animal-based	Signs of pain	86.96	0.1	82.61	0.2	78.26	0.2
	Affiliative behaviours with conspecifics	95.65	0.1	78.26	0.1	78.26	0.2
	Abnormal repetitive behaviours or stereotypies	95.65	0.1	86.96	0.2	95.65	0.2
	Important behaviours observed at expected time and intensity	86.96	0.2	73.91	0.3	78.26	0.3
	Injuries caused by the enclosure or cage	86.96	0.2	73.91	0.2	73.91	0.2
	Injuries caused by other non-human primates	91.3	0.1	78.26	0.2	73.91	0.2
Resource and management-based	Quantity, frequency, variety, and nutritional composition	86.96	0.2	78.26	0.1	78.26	0.2
	Condition and cleanliness of drinker(s)	91.3	0.2	73.91	0.3	73.91	0.2
	Number and cleanliness of feeders	95.65	0.1	78.26	0.2	78.26	0.2
	Animal density	95.65	0.1	100	0.0	100	0.2
	Enclosure or habitat dimensions	86.96	0.1	82.61	0.2	82.61	0.2
	Vertical space: presence of three levels	86.96	0.1	86.96	0.2	86.96	0.2
	Cleanliness of shelter	91.3	0.1	78.26	0.1	78.26	0.2
	Presence of visual barriers between enclosures or cages	86.96	0.1	73.91	0.2	73.91	0.2
	Provision of social enrichment	91.3	0.2	82.61	0.2	82.61	0.2
	Provision of physical enrichment	100	0.0	95.65	0.1	95.65	0.2
	Provision of sensory enrichment	82.61	0.1	86.96	0.2	86.96	0.2
	Provision of cognitive/occupational enrichment	86.96	0.1	91.3	0.2	91.3	0.2
	Availability of cubicles for isolation and safe animal handling	86.96	0.2	86.96	0.2	78.26	0.2
	Social stability	91.3	0.2	73.91	0.2	73.91	0.3

**Table 4 animals-15-02473-t004:** Extended audit-type welfare assessment protocol for *Cebus albifrons* in rescue centres.

Welfare Principle	Welfare Criteria	Welfare Indicator
Good Feeding	Absence of prolonged hunger	Body condition score; Quantity, frequency, variety, and nutritional composition; Number and cleanliness of feeders
	Absence of prolonged thirst	Number of water points in enclosure
Good Housing	Comfort during rest	Sharp edges in the enclosure or habitat; Availability of shelter Cleanliness of shelter; Presence of visual barriers between enclosures or cages; Presence of visual barriers within the enclosure or cage; Availability and type of substrate
	Thermal comfort	Enclosure with access to indoor and outdoor areas
	Ease of movement	Availability of isolation cubicles for safe animal handling; Animal density; Enclosure or habitat dimensions; Vertical space: presence of three levels; Group composition
Good Health	Absence of disease	Evident health alterations; Mortality; Evidence of preventive medicine program
Appropriate Behaviour	Expression of social behaviours	Affiliative behaviours with conspecifics
	Expression of other behaviours	Important behaviours observed at expected times and intensity; Expression of normal species behaviours at unexpected times or intensity; Abnormal repetitive behaviours or stereotypies; Abnormal qualitative behaviours other than stereotypies; Provision of physical enrichment; Provision of sensory enrichment; Provision of cognitive/occupational enrichment
	Good human–animal relationship	Staff training in animal welfare

**Table 5 animals-15-02473-t005:** Daily welfare assessment protocol for *Cebus albifrons* in rescue centres.

Welfare Principle	Welfare Criteria	Welfare Indicator
Good Feeding	Absence of prolonged hunger	Quantity, frequency, variety, and nutritional composition; Number and cleanliness of feeders
Good Housing	Comfort during rest	Cleanliness of shelter
	Ease of movement	Habitat/enclosure dimensions; Vertical space: presence of three levels
Good Health	Absence of disease	Signs of pain
Appropriate Behaviour	Expression of social behaviours	Affiliative behaviours with conspecifics
	Expression of other behaviours	Abnormal repetitive behaviours or stereotypies; Abnormal qualitative behaviours other than stereotypies; Provision of physical enrichment

## Data Availability

The data presented in this study are available on request from the corresponding authors. The data are not publicly available due to confidentiality considerations, as they include responses from expert consultations conducted during the Delphi process. Anonymity and confidentiality were guaranteed to all contributors, in accordance with good research practice.

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
