# Peer review of "Selection of Animal Welfare Indicators for Primates in Rescue Centres Using the Delphi Method: Cebus albifrons as a Case Study"

_animals, 2025, doi:10.3390/ani15172473_

Round 1
Reviewer 1 Report
Comments and Suggestions for Authors
Title: SELECTION OF ANIMAL WELFARE INDICATORS FOR PRIMATES IN RESCUE CENTRES USING THE DELPHI METHOD: CEBUS ALBIFRONS AS A CASE STUDY
The authors have come out new way to assess the animal welfare indicators for Primates at Rescue Centre, based on a case on capuchin monkeys, which is very interesting and useful, given the reason of increasing trend in human-wildlife conflict and the importance of animal welfare in wildlife conservation. By and large the authors done a good job and there is no major lacunae or caveats in the study. However, there are minor issues like modifying, rephrasing, re-ordering, referencing and listing out the welfare issues indicators at the conclusions and giving recommendation for each one of them will make improve the clarity of the manuscript. The following are point-by-point suggestion and the authors can incorporate each one of them.
Point-by-point suggestions
In the title, the letter ‘s’ in centres is not needed.
Line No. 11-12: Wildlife rescue centres care for animals from a wide range of species, creating major challenges for ensuring good welfare conditions.
Suggestion: In the above flow is missing needs to be ensured especially the word from is inappropriate in the location.
The term Delphi is not a common term, please briefly introduce what is Delphi method at the first mention.
Line No. 20-21: but animal-based measures were seen as crucial for assessing actual 20 welfare.
In the above animals-based measures based on what identified as a crucial assessing actual welfare?
Line No. 37: After each round, the participants received feedback on their evaluations to ensure refinement and consensus.
I hope the participants mean experts, if so, please use the term experts, which gives more clarity and a respectful term. Because participants could also be inexperienced
Line No. 42: resource- and management-based measures, which were seen as more practical and easier to implement.
In the above statement resource- and management-based measures are not only measures more practical and easier to implement, but also basically are the root causes of the welfare issues and the animal-based indicators, such as affiliative behaviours and signs of pain are the indicators poor welfare. The same needed to be expressed in the statement.
Further in the above statement instead of outlining resource-and management-based measure, must give a hint of what exactly are they? So that readers who reads the abstract or the summary understand the same.
Line No. 55: Wildlife rescue or rehabilitation centres are establishments designed to temporarily house terrestrial and aquatic wildlife specimens that have been apprehended, confiscated, or voluntarily surrendered for care, evaluation, and treatment.
In the above statement, rescued from the wild also needs to be included.
The first paragraph is merely based on one reference and suggest to include a few more.
Line No. 71: dividing several previously described subspecies. Please remove the term several.
At the end of second paragraph, I suggest the authors to mention what the IUCN Red-List Status of the study species, which is important.
Line No. 78: after their capture and referral to rescue centres. Please replace the term referral with transfer.
Line No. 83: Systematic assessment of animal welfare is a fundamental tool for institutions re-83 sponsible for animal welfare. In this sentence, tool is not needed, may be deleted.
Line No. 98: Variables such as enclosure size, available space, ambient temperature, and access to basic resources such as food, water, and bedding materials (13). In this statement, access to basic resources such as food, water and shelter needs to be introduced first, as they are the fundamental and then mention about the enclosure, weather and social environment etc.
Line No. 99: Meanwhile, management-based indicators relate to operational practices within the institution, including records of health and veterinary care, quality of human-animal interactions, and adherence to standard operating procedures.
As mentioned above comments, the authors when listing many things, need to give priority for the basic or vital steps at first, all others as per their importance, like animal keeper, and veterinary care, visitors, adherence to standard operating procedures and records of health
Line No. 125: Although work has been done to evaluate the welfare of primates in other contexts, such as research 126 laboratories (17–19), zoos and bio parks (20,21), there is currently no system, method, or 127 protocol with valid, reliable, and practical indicators for evaluating the welfare of Cebus 128 albifrons or any other primate species in these institutions.
The above justification for the present study is inappropriate, because the current study was carried out in in rescue centres, while the justification deals about the research laboratories (17–19), zoos and bio parks, where no system, method, or protocol with valid, reliable, and practical indicators for evaluating the welfare of Cebus albifrons or any other primate species in these institutions. Instead, can’t the authors say nil or lack of welfare assessment study in rescue centres? If there are adequate number of studies available in rescue centres, then you should also include rescue centres in your list of institutions mentioned in the line number 125-126.
Figure: Is the questionnaire survey, a part of the figure? If so include it as new step or add along with existing step.
Line No. 147: These experts were then contacted via phone, WhatsApp, and email to thoroughly explain the project's objectives 148 and methodology were thoroughly explained.
The basic reason for contacting the expert is missing in the above statement. These experts were then contacted via phone, WhatsApp, and email to explain the project's objectives, methodology and obtained ??????.
Line No. 160: The participants were asked to evaluate the types of questions, clarity, response options, and time required to complete the 161 questionnaires, allowing for necessary adjustments.
In the above statement participants term is suddenly introduced without any prior mention who are the participants, what for the participants were surveyed? It is confusing. Is participants are not part of the method and if so, why it is appearing in the Delphi?
Line No. 222: 3.1.Expert Demographics 222
Twenty-three participants completed both rounds of the Delphi study; 22 were from 223 Colombia, and one was from Ecuador. Of these, 60.87% (14) work as veterinarians, 13.04% 224 (3) as animal scientists, 8.07% (2) as researchers, and 14.39% (4) in other roles, such as 225 biologists, primatologists, and logistical and technical support staff. All the experts had 226 some form of higher education: 30.43% (7) held an undergraduate degree, 13.04% (3) had 227 a specialisation, 39.13% (9) had a master’s degree, 13.04% (3) held a doctoral degree, and 228 4.34% (1) had a postdoctoral qualification. Most of the participants had experience working with primates (86.96%), either under human care (52.17%) or in both in situ and ex situ 230 conditions (34.78%); of these, 41.83% had more than 10 years of experience. Experts with-231 out primate experience worked in the field of animal welfare. 232
This section needs to be moved to the method section appropriately, as it is part of the objectives. Only results pertaining to the objectives are alone needed to be brought under result section.
The * signs in the axis labelling of figure 2 & 3 need to be mentioned as what they indicate in the figure legend itself and their positioning to be uniform in figure 2 they are after name label, while in the figure 3, they are after the ranking number.
In Table 4. Number of drinkers in enclosure. This is not clear reword this term.
Conclusions heading can also be added with recommendations, so that it not only attracts more readers, but also appropriate for the title concerned.
Further, the topic must list-out the conclusions points briefly and specific recommendation to overcome the specific welfare issue.
Comments on the Quality of English Language
English language can be improved a bit.
Reviewer 2 Report
Comments and Suggestions for Authors
Overall, this survey provides fascinating insight into the consensus among experts on welfare indicators. Although the framing in the introduction suggests that the measures will be validated for the physical markers of poor welfare, they are surveying the consensus on the credibility and practicality of the proposed welfare measures. All the measures may have been validated in a separate study, but this is not explicitly stated here.
Regarding the writing, generally it is clear, but for fluidity and ease of reading the authors should use the active voice throughout. The authors use the passive voice throughout the manuscript, but it is particularly noticeable in the methods section. That section should be in the past tense but not the passive voice. I have a few line comments below.
Introduction
The first paragraph of the introduction should include more citations to support the information throughout. The topic is broad enough that there is more than a single reference available to support the work.
Lines 83-89: The authors include information here that should be cited.
Lines 108-111: It would be worthwhile to talk about the difference across these tests reliably; the first two are more common and probably familiar, but the last warrants a line or two of discussion.
Line 124-132: Rewrite this section in the active voice.
Line 125: considered an ethical and moral responsibility.
Figure 1 is blurry and should be higher resolution.
Table 1: wildlife rescue centre should not be capitalized unless that is the name of a specific centre, and if it is provide more information about it before this point.
The introduction focused on the reliability and validity of these measures, but it is unclear how these two elements were determined. The methods and results suggest it was only a matter of survey participants saying what they think about each measure without the biological assessment to confirm they are a valid measure. It would be worthwhile to highlight the difference here in terms of validity, specifically as a truthful representation of the underlying condition and content validity. If the authors used the term “content validity,” which is what I see the assessment as, throughout, there would be no expectation that these measures have been truly validated against biological measures for this species.
Figure 3: What does the asterisk indicate?
All the tables are aligned in a way that makes it hard to read; consider a left-justified option.
Update the spacing in section 3.4.2.
Line 305: rescue centers does not need to be capitalized.
Line 395-397: The measures do not need to be capitalized.
The discussion focuses heavily on participation drop off, where it may be more advantageous to focus on the measure and what they mean for the welfare of the primates. The discussion should follow the reframing needed in the introduction.
